# Are *COL22A1* Gene Polymorphisms rs11784270 and rs6577958 Associated with Susceptibility to a Non-Contact Anterior Cruciate Ligament Injury in Polish Athletes?

**DOI:** 10.3390/ijerph20010515

**Published:** 2022-12-28

**Authors:** Zhuo Sun, Paweł Cięszczyk, Ewelina Lulińska, Magdalena Dzitkowska-Zabielska, Monika Johne, Kinga Humińska-Lisowska, Monika Michałowska-Sawczyn, Krzysztof Ficek, Agata Leońska-Duniec, Andrzej Mastalerz, Arkadiusz Janczyk, Sawczuk Marek

**Affiliations:** 1Faculty of Physical Education, Gdansk University of Physical Education and Sport, 80-336 Gdansk, Poland; 2Faculty of Physical Education, Jozef Pilsudski University of Physical Education in Warsaw, 00-809 Warsaw, Poland; 3Faculty of Physiotherapy, The Jerzy Kukuczka Academy of Physical Education in Katowice, 40-065 Katowice, Poland; 4Center for Physiotherapy, Military Unit No. 6021, 01-001 Warsaw, Poland; 5Institute of Physical Culture Sciences, University of Szczecin, 70-453 Szczecin, Poland

**Keywords:** *COL22A1* gene, injury, ACL, athletes

## Abstract

Understanding the risk factors and etiology of ACL ruptures (anterior cruciate ligament) is crucial due to the injury’s high occurrence, significant financial cost to the healthcare sector, and clinical consequences. In this study, we investigated the hypothesis that rs11784270 A/C and rs6577958 C/T SNPs (single gene polymorphism) within *COL22A1* are associated with ACL ruptures (ACLR) in Polish soccer players. Methods: 228 athletes with ACLR (157 male, age 26 ± 4, 71 female, age 26 ± 6) and 202 control athletes (117 male, age 26 ± 6, 85 female, age 29 ± 2) engaged in the study. The buccal cell swabs were genotyped using TaqMan^®^ pre-designed SNP genotyping assays, following the manufacturer’s recommendations. The R program and SNPassoc package were used to determine the genotype and allele frequency distributions under the various inheritance models (co-dominant, dominant, recessive, and over-dominant). Further, *p*-values of <0.05 were considered statistically significant. We found no association between the analyzed polymorphisms and the risk of non-contact ACL ruptures in any of the studied models. Although the genetic variants investigated in this study were not associated with the risk of non-contact ACL ruptures, we assumed that the *COL22A1* gene remains a candidate for further investigations in musculoskeletal injuries.

## 1. Introduction

In sports medicine, musculoskeletal injuries (MSI) are one of the main health issues. The occurrence of the types and locations of injuries, e.g., joints, muscles, tendons, and ligaments, varies depending on the sport type and ranges from 5% to 60% [1,2].

The anterior cruciate ligament rupture (ACLR) is a frequent and serious knee injury in athletics, usually occurring without any impact [3,4]. ACL reconstruction is the preferable treatment. A total of 65% of patients typically resume their level of athletic involvement before surgery, and 55% do so at a competitive level [5,6].

Understanding the risk factors and etiology of ACL ruptures is important due to the injury’s high occurrence, the significant financial cost to the healthcare sector, and the clinical consequences.

Many internal and external risk factors can result in ACL damage. Anatomical variances, sex, and genetic variability are examples of intrinsic factors [7,8]. It has been suggested that due to differences in the DNA sequence, some people may be more susceptible to ACL damage than others [9,10,11,12].

Tendons and ligaments, such as the ACL, are strands of dense regular bundles of connective tissue comprising multiple collagen fibers (such as I, III-VI, and XII). The bundles are surrounded by dense irregular connective tissue sheaths as well as various non-collagen particles, including proteoglycans and glycoproteins [13]. Collagen fibers are encoded by several genes. Collagen-type XXII alpha 1 chain, a quantitatively minor collagen of the fibril-associated collagen family with interrupted triple helices, is encoded by the *COL22A1* gene (FACITs). The biological function of this protein is not fully understood. However, it seems to support the strengthening of skeletal muscle attachments and the stabilization of myotendinous junctions (MTJ) during contractile activity [14].

In zebrafish, a loss-of-function mutation in the collagen-type XXII a-1 chain gene causes a phenotype that resembles muscular dystrophy and reduced force production. This implies that collagen is involved in the maintenance of the functional efficacy and stabilization of the MTJ [15].

In one of the studies, two SNPs in the *COL22A1* gene, namely rs11784270 and rs6577958 (the A and T alleles, respectively), were significantly associated with non-contact muscle damage in Japanese athletes [16].

Given that both the rs11784270 A/C and rs6577958 C/T SNPs located within *COL22A1* are not well described and their connection with ACL injuries is not clearly defined, we decided to test the hypothesis that they are associated with ACL susceptibility to injury in Polish athletes.

## 2. Materials and Methods

### 2.1. Participants

The study involved 430 unrelated, self-reported Caucasians recruited between the years 2009 and 2016. The anterior cruciate ligament rupture (ACLR) case group included 228 people: 157 men and 71 women. The inclusion criterion was those surgically diagnosed with primary ACLR and eligible for ligament reconstruction. The 228 subjects of the ACLR group who suffered injuries did so without physical contact. There were 202 participants in the control group (CON), 117 men and 85 women, all of whom appeared to be in good health and had never experienced an ACL injury. The ACLR males all played soccer in Poland’s first, second, and third-level leagues and trained 11 to 14 h per week on average (mean time: 11.9 ± 1.4). They had an average age of 26 years and 4 months. The ACLR females (mean age: 26 ± 6 years) were soccer players from the Polish first and second-level leagues (trained 10–12 h per week; mean time: 11.1 ± 0.6). The male control group was composed of healthy, physically active adults with a mean age of 26 years and had similar amounts of exposure to sports (identical levels of training and competition intensity). The female control subjects, who self-reported being physically active for a minimum of 7 h per week (mean time 9.2 ± 1.4), were chosen from sports clubs and wellness facilities (mean age: 29 ± 2 years). Information about the study participants is summarized in Table 1.

### 2.2. Ethical Approval

The study’s methods were approved by both the Bioethics Committee for Clinical Research of the Regional Medical Society in Gdansk (approval number KB 8/16) and the Ethics Committee of the Pomeranian Medical University in Szczecin, Poland (approval number 09/KB/IV/2011). The protocols adhered to the rules of the World Medical Association Declaration of Helsinki. An information sheet outlining the specifics of the study, including its goal, the procedures involved, as well as any risks and advantages of participation, was given to each participant. Anonymity and confidentiality were preserved in the study. All participants gave their written informed consent. The Strengthening the Publishing of Genetic Association Studies (STREGA) Statement, which establishes a set of guiding standards for reporting the findings of genetic association studies, was followed in the conduct of this case-control study [17].

### 2.3. Genetic Analyses

The buccal cells were collected using Copan FLOQSwabs (Copan Diagnostics, Inc., Murrieta, CA, USA) during post-surgery control. DNA was extracted using a GenElute Mammalian Genomic DNA Miniprep Kit (Sigma, Taufkirchen, Germany) in agreement with the manufacturer’s instructions. On a StepOne Real-Time Polymerase Chain Reaction (RT-PCR) instrument (Applied Biosystems, Bedford, MA, USA), all samples were genotyped in duplicate using the TaqMan^®^ pre-designed SNP genotyping assays (C_176045196_10 for the *COL22A1* rs11784270 and C_29400116_10 for the *COL22A1* rs6577958, Applied Biosystems, Bedford, MA, USA) according to the protocol [12].

### 2.4. Statistical Analysis

Statistical analyses were performed using the R programming environment and R package (version 3.4.0, The R Foundation for Statistical Computing, https://cran.r-project.org, accessed on 30 August 2022). Genotype and allele frequency distributions were calculated using four models of inheritance (co-dominant, dominant, recessive, and over-dominant) and the SNPassoc package 2.1-0. Further, *p*-values of <0.05 were used as a cut-off for significance.

## 3. Results

Table 2 and Table 3 demonstrate the genotypes and allele frequency distributions for the two analyzed *COL22A1* SNPs: rs11784270 (Table 2) and rs6577958 (Table 3). In order to avoid bias in the haplotype frequency estimates, none of the polymorphisms in the co-dominant, dominant, recessive, and over-dominant models were associated with the risk of non-contact ACL ruptures. The genotype frequencies among groups are illustrated in Figure 1.

## 4. Discussion

This study aimed to investigate the association polymorphism variants located within *COL22A1* with ACL injuries. The main findings of this genetic association study show (i) no independent associations between the rs11784270 A/C and rs6577958 C/T polymorphisms and non-contact ACL ruptures; (ii) none of the polymorphisms in the co-dominant, dominant, recessive, and over-dominant models were associated with the risk of non-contact ACL ruptures.

The *COL22A1* gene encodes COLXXII, FACIT-type collagen, and its biological role has not been fully understood. This protein has been described as a unique tissue junction element found predominantly in transitional tissue zones, e.g., the MTJ of the skeletal muscle, the heart, the articular cartilage-synovial fluid junction, and the junction between the hair follicle and dermis [18,19]. Zebrafish COLXXII depletion resulted in muscular dystrophy, probably due to a disrupted myotendinous junction [15]. In addition, adult homozygous mutants had a higher frequency of cerebral hemorrhages due to increased vascular permeability [20]. It has recently been shown that *COL22A1* is involved in mice bone remodeling and is expressed in bone-forming osteoblasts, but not bone-resorbing osteoclasts—*COL22A1-*deficient mice displayed trabecular osteopenia as well as a significantly increased osteoclast number [19]. In humans, *COL22A1* genetic variants have been linked to a higher risk of aneurysms and are being investigated for their potential contribution to musculoskeletal soft tissue injuries. Specifically, it has been shown that SNPs associated with different mRNA expression levels of the collagen-type XXII gene (rs11784270 and rs6577958) are related to athlete susceptibility to muscle injuries. The high expression of A (rs11784270) and T (rs6577958) alleles were strongly linked to muscle damage in athletes when analyzed using additive genetic models. According to the authors, a high expression of *COL22A1* at the MTJ is linked to muscle injury risk in athletes [16].

Our genetic association study set out to determine the possible interaction between these two SNPs within the *COL22A1* gene (rs11784270 and rs6577958) and non-contact ACL ruptures in a Polish cohort. However, under the co-dominant, dominant, recessive, and over-dominant models, we found no independent associations between the rs11784270 and rs6577958 polymorphisms and non-contact ACL tears.

Since the available literature concerning the role of COLXXII in ACL formation and remodeling is unclear, it is difficult to determine whether the *COL22A1* polymorphisms are associated with injury risk to the ACL.

We chose a genetic marker for the study based on the assumption that tendons and ligaments have similar basic structures and gene expression patterns of their major cell types [13].

Ligaments are bands of connective tissue made of collagenous fibers that connect bones to other bones to form joints, while tendons connect bones to muscles. The key factors affecting the mechanical characteristics of ligaments and their subsequent structural integrity have been demonstrated to be the collagen content and cross-linking between collagen fibers [21,22,23]. In healthy ligaments, collagen replacement occurs. This manifests in collagen fibrils’ cross-linking maturation and elevated tensile strength [23,24]. The high collagen turnover in such ligaments is maintained in part by mechanical incentives [25,26]. For instance, exercise has been demonstrated to cause an increase in the synthesis of ligament collagen [24,27].

To date, several studies, including our own, have shown that a variation in collagen coding genes contributes to ACL injury risk. These genes were *COL1A1, COL3A1, COL5A1*, and *COL12A1,* which were associated with ACL rupture [28,29,30,31,32,33].

The regulation of expression and the appropriate proportion of collagen fibers that build ligaments and tendons are crucial for their integrity. For example, a higher expression of the main collagenous component of the ligaments and tendons, *COL1A1,* might raise the possibility of ACL injury by disrupting the structural integrity of collagen fibers [28,34]. The most common *COL1A1* variant, rs1800012 (+1245G/T, Sp1), results in the G allele’s substitution with the T allele within the gene’s first intron. The rs1800012 SNP leads to an increase in the levels of type I collagen in α1 chains. In turn, this results in a disproportion in the ratio between the α1 and α2 chains (2:1) by increasing the binding affinity of RNA polymerase II [12,35,36,37].

A higher *COL22A1* expression caused by the A and T alleles of rs11784270 and rs6577958, respectively, has been associated with muscle injury in athletes. It has been postulated that high levels of *COL22A1* expression at the MTJ alter the MTJ properties [16]. The rs11784270 and rs679620 variants are detected within the intron of *COL22A1*. According to the GTEx Portal (https://www.gtexportal.org/home, accessed on 30 August 2022), the A and T alleles have higher transcriptional activity than the C and C alleles of the rs11784270 and rs679620, respectively [16]. However, there is currently an insufficient understanding of the molecular mechanisms that determine the high expression of the *COL22A1* A and C alleles. Therefore, efforts should be made to better understand the process underlying *COL22A1* expression alteration. Despite the fact that the SNPs analyzed in the present study were not associated with the risk of non-contact ACL rupture, we assumed that the *COL22A1* gene remains an important biological candidate for further interrogation of musculoskeletal injuries based on the findings of earlier case-control genetic association studies [16,28].

## 5. Conclusions

Despite the fact that the genetic variations examined in this study were not linked to the risk of non-contact ACL injury, we inferred from earlier case-control genetic association studies that the *COL22A1* gene is still a crucial biological candidate for further investigation in musculoskeletal injuries.

## Figures and Tables

**Figure 1 ijerph-20-00515-f001:**
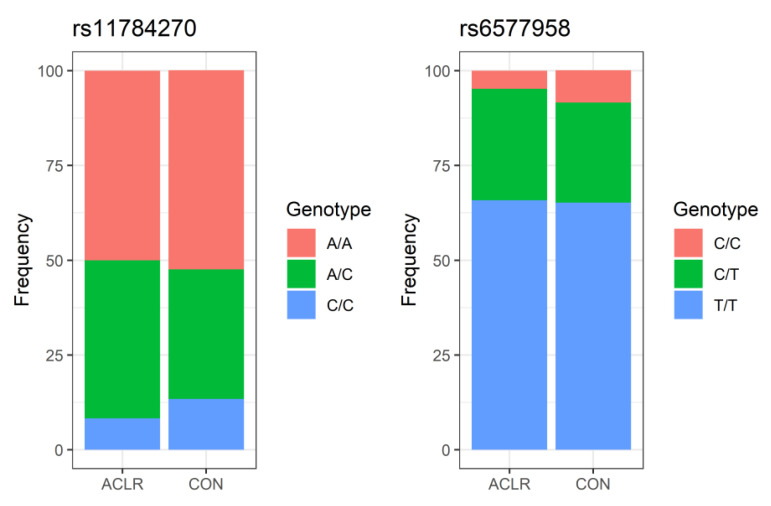
Genotypes frequency in ACLR and control groups.

**Table 1 ijerph-20-00515-t001:** Characteristics of the study participants.

	*N*	Sex	Age	Time of Training (h/per Week)
Case group: athletes with ACL rapture	228	157 men	26 ± 4	11.9 ± 1.4
		71 women	26 ± 6	11.1 ± 0.6
Control group: athletes without ACL rapture	202	117 men	26 ± 6	11.2 ± 1.2
		85 women	29 ± 2	9.2 ± 1.4

**Table 2 ijerph-20-00515-t002:** Analysis of the relationship between non-contact ACL rupture and the *COL22A1* gene rs11784270 A/C polymorphism.

Model		CON (*n* = 202)	%	ACLR (*n* = 228)	%	OR	95% CI		*p*-Value
Co-dominant	A/A	106	52.5	114	50.0	1.00			0.119
	A/C	69	34.2	95	41.7	1.28	0.85	1.92	
	C/C	27	13.4	19	8.3	0.65	0.34	1.25	
Dominant	A/A	106	52.5	114	50.0	1.00			0.608
	A/C-C/C	96	47.5	114	50.0	1.10	0.76	1.61	
Recessive	A/A-A/C	175	86.6	209	91.7	1.00			0.092
	C/C	27	13.4	19	8.3	0.59	0.32	1.10	
Over-dominant	A/A-C/C	133	65.8	133	58.3	1.00			0.109
	A/C	69	34.2	95	41.7	1.38	0.93	2.04	
Allele	A	281	69.6	323	70.8	1.00			0.682
	C	123	31.4	133	29.2	1.06	0.79	1.42	

ACLR—anterior cruciate ligament rupture, CON—control, OR—odds ratio, 95% CI—confidence intervals.

**Table 3 ijerph-20-00515-t003:** Association analysis of the *COL22A1* gene rs6577958 T/C polymorphism with non-contact ACL rupture.

Model		CON (*n* = 202)	%	ACLR (*n* = 228)	%	OR	95% CI		*p*-Value
Co-dominant	T/T	131	65.2	150	65.8	1.00			0.284
	C/T	53	26.4	67	29.4	1.10	0.72	1.70	
	C/C	17	8.5	11	4.8	0.57	0.26	1.25	
Dominant	T/T	131	65.2	150	65.8	1.00			0.894
	C/T-C/C	70	34.8	78	34.2	0.97	0.65	1.45	
Recessive	T/T-C/T	184	91.5	217	95.2				0.128
	C/C	17	8.5	11	4.8	0.55	0.25	1.20	
Over-dominant	T/T-C/C	148	73.6	161	70.6				0.487
	C/T	53	26.4	67	29.4	1.16	0.76	0.76	
Allele	T	315	78.4	367	80.5	1.00			
	C	87	21.6	89	19.5	1.14	0.82	1.59	0.442

ACLR—anterior cruciate ligament rupture, CON—control, OR—odds ratio, 95% CI—confidence intervals.

## Data Availability

Not applicable.

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
