# Peer review of "Are COL22A1 Gene Polymorphisms rs11784270 and rs6577958 Associated with Susceptibility to a Non-Contact Anterior Cruciate Ligament Injury in Polish Athletes?"

_ijerph, 2022, doi:10.3390/ijerph20010515_

Round 1
Reviewer 1 Report
This is an impressive study,but it has some serious drawbacks.
1. Why were the gene polymorphisms described in this article chosen to assess their susceptibility to non-contact ACL injury even though the previous studies (ref 13,14)show only their vulnerability in damaging the muscle or MTJ?
2. Why only non-contact ACL injury was chosen in this study ? Please give your inputs.
3.Why contact ACL injury excluded from the study ? Please provide valid explanation to exclude the above group ?
4. Whether the female control subjects chosen from sports clubs and wellness facilities were subjected to the same level of intense pivoting sports as in the study group ??
5. In the conclusion part the authors have stated that COL22A1 gene is still a crucial biological candidate for further investigation in musculoskeletal injuries. Please explain why the authors have taken only ACL injury rather than taking a relatively bigger group of musculoskeletal injuries ?
6. Please explain how, if the authors were to establish the association between the gene polymorphisms and ACL injury, this would influence the management policy.?
7.Based on question 6, if the management policy changes, will that be a paradigm shift in the management of ACL injuries?
We look forward to hearing from you soon.
Thank you.
Author Response
Thank you very much for your comments and we send our answers below:
This is an impressive study,but it has some serious drawbacks.
- Why were the gene polymorphisms described in this article chosen to assess their susceptibility to non-contact ACL injury even though the previous studies (ref 13,14)show only their vulnerability in damaging the muscle or MTJ?
Response: We thank reviewer 1 for this useful comment. We would like to explain our choice.
Collagen XXII (Col XXII), which shows unique localization at tissue junctions in muscle, tendon, heart, articular cartilage, and skin, seemed to be a new marker in muscle dysfunction. To date, research on COL22A1 has mainly been conducted in animal models. In the literature, there is only one publication in which the polymorphisms in COL22A1 are examined among athletes. Since the collagen encoded by the COL22A1 gene is involved in the maintenance of MJT strength, it seems reasonable to investigate the impact of these polymorphisms in other injuries/diseases.
This is a pioneering approach and adds new cognitive value to the existing state of knowledge.
- Why only non-contact ACL injury was chosen in this study? Please give your inputs.
Response: We thank reviewer 1 for this comment.
Non-contact injuries account for the majority of ACL injuries. It is a well-known and well-described research model with a homogeneous etiology. Determining the genetic background in terms of mixed etiology of injuries (non-contact and contact, caused by an external force) would have yielded inconclusive result. The possibility of comparing the obtained data with the results of other scientists in the same research model was also an important factor.
- Why contact ACL injury excluded from the study? Please provide valid explanation to exclude the above group?
Response: We thank reviewer 1 for this comment.
The answer to this question is partly explained above. Only non-contact injuries were included in the study to simplify the model (the aetiology of the injuries was the same for all participants, and additional factors related to the occurrence of the injury were excluded).
- Whether the female control subjects chosen from sports clubs and wellness facilities were subjected to the same level of intense pivoting sports as in the study group??
Response: We thank reviewer 1 for this comment.
The female control group consisted mainly of soccer players. However, 7 women were recruited from sports and wellness clubs and had comparable levels of sports exposure to the study group (similar volume and intensity of training).
- In the conclusion part the authors have stated that COL22A1 gene is still a crucial biological candidate for further investigation in musculoskeletal injuries. Please explain why the authors have taken only ACL injury rather than taking a relatively bigger group of musculoskeletal injuries ?
Response: We thank reviewer 1 for this comment.
As mentioned above the noncontact ACL injury is a plausible, well-characterized, and homogenous model, but this article is the first study and should certainly be expanded in the future to include other types of injury.
- Please explain how, if the authors were to establish the association between the gene polymorphisms and ACL injury, this would influence the management policy?
Response: We thank reviewer 1 for this comment.
The genetic variants tested in our study were not associated with the risk of non-contact ACL rupture. We hypothesized that the COL22A1 gene remains a candidate for further investigation in other musculoskeletal injuries. However, even a positive correlation between COL22A1 polymorphisms and ACL injuries will not influence management policy. The recommended treatment is surgical intervention. By further investigating genetic variation in the genes encoding the proteins that build muscles, tendons and ligaments and determining their mechanical strength, we hoped to create a genetic risk model to determine the predisposition of athletes to ACL injury. GRS will allow us to estimate the likelihood of injury and, with the right training and physiotherapist care, minimize the injury’s risk.
- Based on question 6, if the management policy changes, will that be a paradigm shift in the management of ACL injuries?
Response: We thank reviewer 1 for this comment.
At this point in time, our state of knowledge and the results obtained do not allow us to discuss this further.
Reviewer 2 Report
This was reviewed with interest. I have some important questions regarding the manuscript along with some comments to hopefully clarify some determining factors.
Abstract: data about the subjects will be helpful for the readers (age, level, etc).
Introduction:
- Line 40 and 41: symbols
- Line 56: Space after dot.
- Line 58: reference would be convenient.
- In general, it is recommended to review more recent evidence for certain statements in this section.
Materials and Methods:
- A table with the main data of the sample is required.
- The difference in the competitive level of the sample may not represent an important bias?
- A more in-depth description of the inclusion criteria is necessary.
- More detailed information on the sampling process is required for subsequent analysis.
Results:
- The size of the effect should be reflected
- The inclusion of an image would help to understand the results.
Discussion:
- This section should begin by clarifying whether the stated objective has been met.
- Since it is difficult to determine whether COL22A1 polymorphisms are associated with injury risk to the ACL, it is necessary to reinforce the impact and contributions of this research in the current evidence.
Author Response
Thank you very much for your comments Reviewer 2 and we send our answers below:
1) Abstract: data about the subjects will be helpful for the readers (age, level, etc).
Response: We thank the reviewer for this suggestion. We have added information about the sex and age of the participants, the size of ACLR, and control group.
2) Introduction:
- Line 40 and 41: symbols
- Line 56: Space after dot.
- Line 58: reference would be convenient.
- In general, it is recommended to review more recent evidence for certain statements in this section.
Response: We thank the reviewer for this suggestion. We have changed this accordingly and added a more recent publication. Although some of the articles are not the most recent, the knowledge is still up to date and cited in many publications.
3) Materials and Methods:
- A table with the main data of the sample is required.
- The difference in the competitive level of the sample may not represent an important bias?
- A more in-depth description of the inclusion criteria is necessary.
- More detailed information on the sampling process is required for subsequent analysis.
Response We thank the reviewer for this suggestion. We have added a table (Table 1) summarizing participants’ data. We considered possible influence differences in the competitive level of the sample however, as all participants in the study had similar volume and intensity of training, we decided to group them together. In addition, a division according to the difference in the level of competitors would spoil them into small groups, and this would make statistical analysis impossible. The main inclusion criteria for the ACLR group was medically confirmed primary ACLR, eligible for ligament reconstruction. As almost all the participants were soccer players, they displayed similar activity patterns, and the exclusion criteria for the ACLR and control group was previous ACLR in medical history.
We added more information about sample collection and the reference article where the genotyping protocol is described.
4) Results:
- The size of the effect should be reflected
- The inclusion of an image would help to understand the results.
Response: We thank the reviewer for this suggestion. The odds ratio and 95% CI reflect the size effect and it has been described in the results section. We have added a figure (Figure 1) with the genotype frequency in both groups.
5) This section should begin by clarifying whether the stated objective has been met.
- Since it is difficult to determine whether COL22A1 polymorphisms are associated with injury risk to the ACL, it is necessary to reinforce the impact and contributions of this research in the current evidence.
Response: We have added information about the results at the beginning of the discussion section.
We thank the reviewer for useful this suggestion. The best option in this range will be conducting more research on larger groups. The second approach is to study genomic interactions.
Round 2
Reviewer 1 Report
Thank you for your well explained notes to my queries.
Reviewer 2 Report
Congratulations for the big improvements done